# Modeling and Predicting Urban Expansion in South Korea Using Explainable Artificial Intelligence (XAI) Model

Minjun Kim and Geunhan Kim *

Department of Environmental Planning, Korea Environment Institute, Sejong 30147, Korea
* Correspondence: ghkim@kei.re.kr; Tel.: +82-044-415-7752

**Abstract:** Over the past few decades, most cities worldwide have experienced a rapid expansion with unprecedented population growth and industrialization. Currently, half of the world's population is living in urban areas, which only account for less than 1% of the Earth. A rapid and unplanned urban expansion, however, has also resulted in serious challenges to sustainable development of the cities, such as traffic congestion and loss of natural environment and open spaces. This study aims at modeling and predicting the expansion of urban areas in South Korea by utilizing an explainable artificial intelligence (XAI) model. To this end, the study utilized the land-cover maps in 2007 and 2019, as well as several socioeconomic, physical, and environmental attributes. The findings of this study suggest that the urban expansion tends to be promoted when a certain area is close to economically developed area with gentle topography. In addition, the existence of mountainous area and legislative regulations on land use were found to significantly reduce the possibility of urban expansion. Compared to previous studies, this study is novel in that it captures the relative importance of various influencing factors in predicting the urban expansion by integrating the XGBoost model and SHAP values.

**Keywords:** urban expansion; explainable artificial intelligence (XAI); land cover; remote sensing

## 1. Introduction

Over the past few decades, most cities worldwide have experienced a rapid expansion with unprecedented population growth and industrialization [1,2]. Cities can be attractive to people as they guarantee a convenient and stable quality of life, with various opportunities including employment, education, and culture [3]. As a result, about 55% of the world's population is currently living in urban areas, which only account for less than 1% of the Earth's land area [4,5].

A rapid and unplanned urban expansion, however, has also resulted in serious challenges to sustainable development of the cities. Urban sprawl and suburbanization has increased the burden of infrastructure and traffic congestion [6,7], and a loss of natural environment and open spaces within the cities has reduced carbon sinks and biodiversity [8,9], while intensifying air pollution and global warming [10,11].

In order to maximize the benefits of urbanization while supplementing its adverse effects, it is essential to predict and control the expansion of urban areas. Accordingly, scholars and practitioners in the urban planning field have developed various methods for modeling the urban growth. Remote sensing data, including satellite images, are utilized in a majority of urban growth models, as they contain a wide range of land use/land cover (LULC) information at the same time [12].

In early studies, the mathematical models, including land-use transportation (LUT) models [13,14], agent-based models (ABMs) [15,16], and cellular automata (CA)-based models [17], were widely used in predicting urban expansion. However, as they assumed that urban areas are spatially homogeneous, those models have difficulties in reflecting the socioeconomic and physical variations within the city [18].

To overcome the limitations of the conventional approach, the machine learning (ML) and artificial intelligence (AI)-based techniques have recently been adopted for urban expansion modeling [19]. The decision tree [20], random forest [21,22], support vector machine [23,24], and artificial neural networks [5,25] are some of the most widely used models in existing research. Although the implementing mechanisms and algorithm structures of each model are different, they have a common strength in developing highly accurate urban growth models by collecting and training large amounts of LULC and physical/social characteristics in the city [26]. However, as prediction accuracy increases, the difficulties in understanding the relationships among variables has been pointed out as one of the limitations of these black-box urban growth models [27].

More recently, an explainable artificial intelligence (XAI) framework has been highlighted among researchers to overcome the weakness of the aforementioned ML and AI models [28]. While there is no clear definition of the concept yet, the XAI aims to increase interpretability and explainability of AI models [29]. In this regard, XAI models require an additional explainable algorithm, such as Shapley Additive exPlanations (SHAP), to explain how and why an AI model achieves a specific result [30].

The purpose of the study lies in modeling and predicting the expansion of urban areas in South Korea by utilizing an XAI model. In addition, we aimed at examining the relative importance of several built-environment factors on the urbanization. To this end, the study utilized the land-cover maps in 2007 and 2019, as well as several socioeconomic, physical, and environmental attributes. The following section describes the materials and methods used in the study; then, we summarize and discuss the findings of the study, and conclude with several recommendations for future studies.

## 2. Materials and Methods

### 2.1. Study Area

The spatial extent of the study was the entire land area covered by the South Korea territory, including Jeju Island. As of 2019, the study area consists of 1 special, 1 self-governing, 6 metropolitan cities, and 8 provinces, with a total area of approximately 118,118.94 km$^2$ (Figure 1). The temporal extent of the study was the expansion of urban areas from 2007 to 2019. For each year, the medium-classified land-cover maps were utilized provided by Ministry of Environment (https://egis.me.go.kr/ (accessed on 1 August 2022)).

South Korea has been one of the fastest-growing countries in the world over the past few decades. From 1980 to 2020, the country's population and gross domestic product (GDP) has increased by 39% and 2393%, respectively, while the land area increased by only 1.3% (https://kosis.kr/ (accessed on 1 August 2022)). As a result, the proportion of urban areas in Korea has dramatically increased from 2.1% in 1980 to 16.7% in 2020, and the population living in urban areas has also doubled during the same period. As of 2020, more than 90% of the nation's population lives in urban areas, and a half of them are concentrated in the capital city, Seoul, and the nearby metropolitan areas [31].

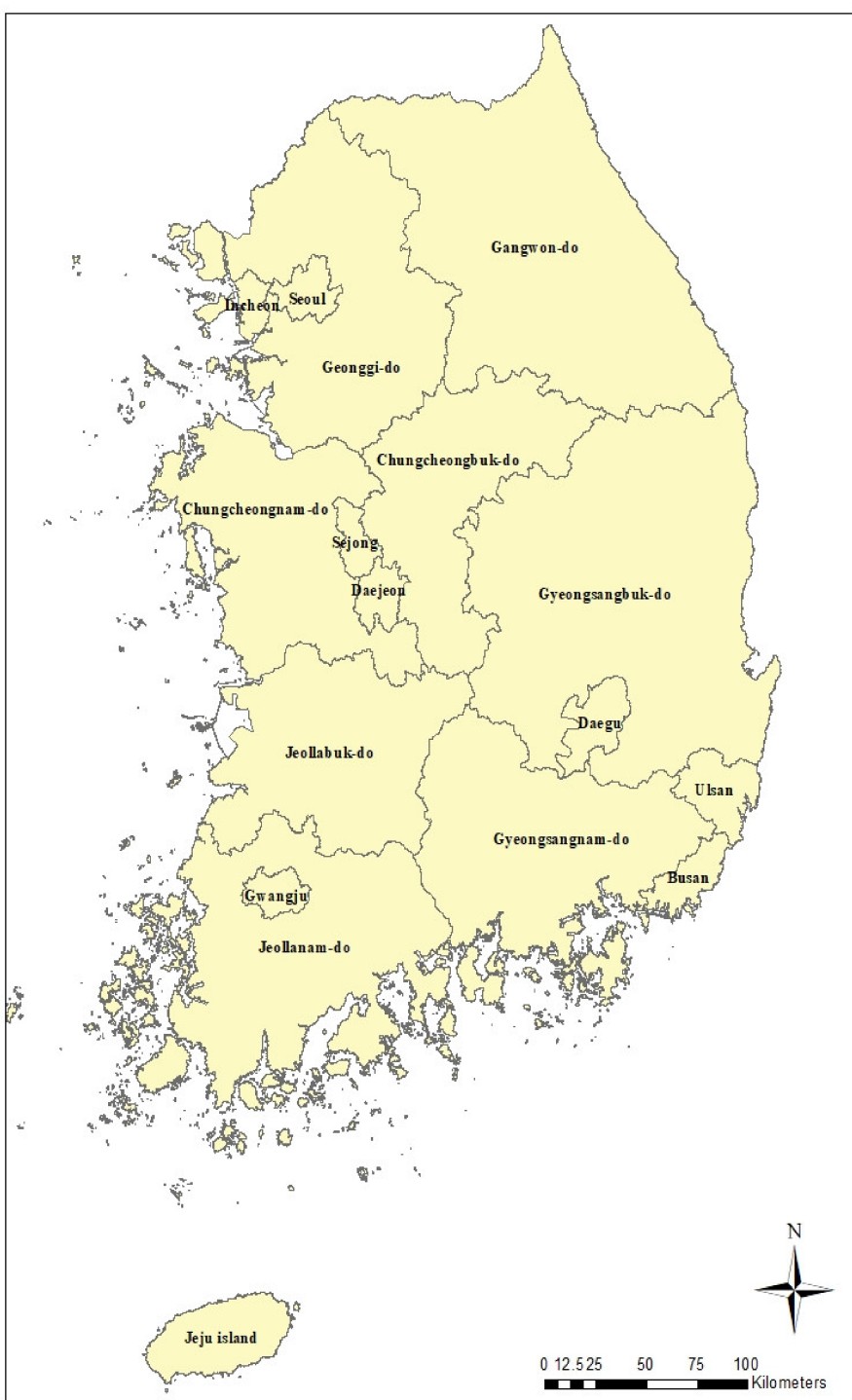

**Figure 1.** Spatial extent of the study.

*2.2. Data*

To analyze and predict the urban expansion in Korea, we constructed national-wide land-cover, topographic, environmental, and socioeconomic feature data for 2007 and 2019, respectively. Table 1 describes the dependent and independent variables used in the study and their sources. For the analysis, all variables were adjusted to a raster grid with a resolution of 10 m.

**Table 1.** Description of variables used in the study.

| Data | | | Source (Year) |
|---|---|---|---|
| **Dependent Variable** | Dummy variables for urbanization from 2007 to 2019 (0: non-urbanized area, 1: urbanized area) | | Land Cover Map (2007 and 2019) |
| **Independent Variable** | Socioeconomic features | Population density | SGIS and KOSIS (2007 and 2019) |
| | | GRDP per capita | |
| | Topographic features | Elevation | Digital Elevation Map (2007 and 2019) |
| | | Slope | |
| | Land-cover features (distance to:) | Residential area | Land Cover Map (2007 and 2019) |
| | | Commercial area | |
| | | Industrial area | |
| | | Transportation area | |
| | | Agricultural area | |
| | | Forest area | |
| | | Grassland area | |
| | | Wetland area | |
| | | Bare land area | |
| | | Water area | |
| | Environmental features | Ecological ECVAM grade | ECVAM (2007 and 2019) |
| | | Legislative ECVAM grade | |

2.2.1. Dependent Variable

In this study, the dependent variable was composed of dummy variables that indicate whether a certain region had been urbanized or not from 2007 to 2019. To this end, we first classified residential, commercial, and industrial areas on the land-cover map as urban areas, otherwise as non-urban areas. We then defined a raster cell which changed from a non-urban area in 2007 to an urban area in 2019 as an 'urbanized area'. A raster cell that remained as a non-urban area in both 2007 and 2019, on the other hand, was defined as a 'non-urbanized area'. Raster cells that had already been classified as urban areas in 2007 were excluded from this study, since they do not correspond with the urban expansion. Table 2 summarizes the classification of urbanized and non-urbanized area in the study.

**Table 2.** Classification of urbanized and non-urbanized area.

| | | 2019 | |
|---|---|---|---|
| | | Urban Area (Residential, Commercial, Industrial Area) | Non-Urban Area |
| **2007** | Urban Area (Residential, Commercial, Industrial Area) | - | - |
| | Non-Urban Area | 1 (Urbanized) | 0 (Non-urbanized) |

2.2.2. Independent Variable

Independent variables of the study consist of (1) socioeconomic, (2) topographic, (3) land-cover, and (4) environmental features. Socioeconomic and environmental features were constructed based on national statistics, and topographic, land-cover features were derived from remotely sensed data (Figure 2).

First, socioeconomic features include population density and gross regional domestic product (GRDP) per capita. Each feature was derived from Statistical Geographic Information Service (SGIS, https://sgis.kostat.go.kr/ (accessed on 1 August 2022)) and Korean Statistical Information Service (KOSIS, https://kosis.kr/ (accessed on 1 August 2022)), respectively. The spatial unit of population density was the census block group level, while that of GRDP per capita was the county level. In this study, raster cells that were located within each census boundary were assigned to corresponding values in 2007 and 2019.

Second, this study adopted the elevation and slope of each raster cell as topographic features. To calculate those, we utilized 10 m digital elevation model (DEM) data provided by the National Spatial Data Infrastructure Portal (http://www.nsdi.go.kr/ (accessed on 1 August 2022)) and the Surface tool in ArcGIS software. As a land-cover feature, we calculated the closest distance from a certain raster cell to other land-cover types. To this end, the Euclidean distance tool in ArcGIS was applied to residential, commercial, industrial, transportation, agricultural, forest, grassland, wetland, bare land, and water area on land-cover maps in 2007 and 2019.

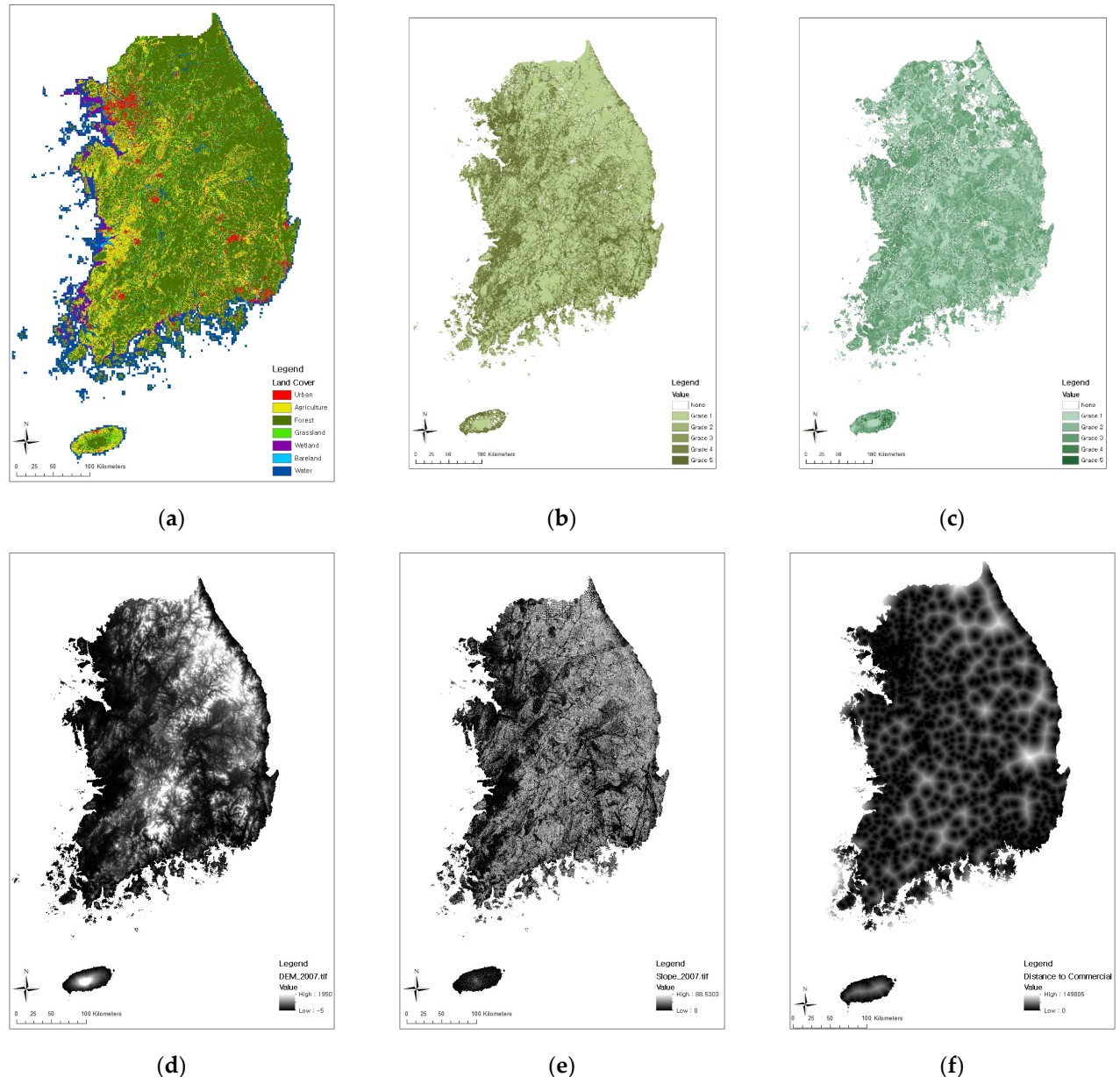

**Figure 2.** *Cont.*

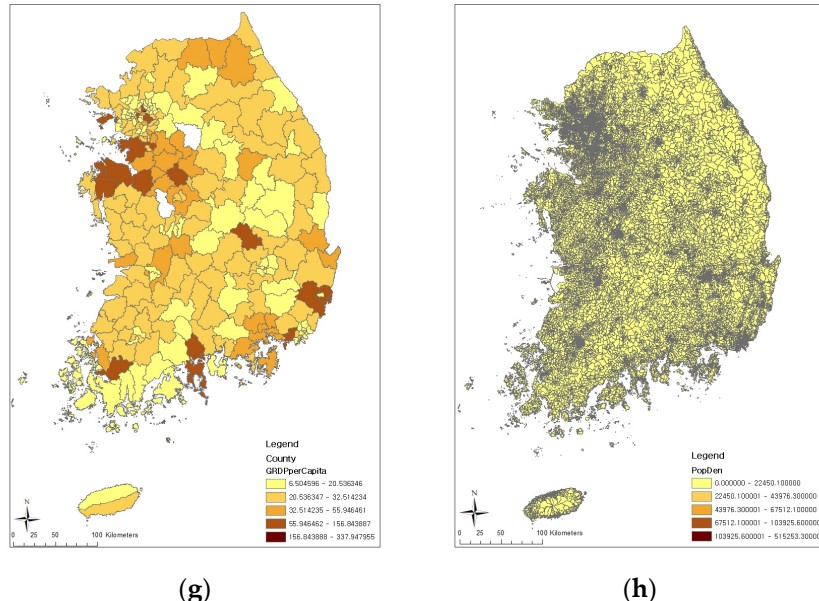

(**g**)                                     (**h**)

**Figure 2.** Variables used in the study. (**a**) Land cover, (**b**) ECVAM ecological grade, (**c**) ECVAM legislative grade, (**d**) elevation, (**e**) slope, (**f**) distance to land cover, (**g**) GRDP per capita, (**h**) population density.

Last, the Environmental Conservation Value Assessment Map (ECVAM) provided by the Ministry of Environment (https://ecvam.neins.go.kr/ (accessed on 1 August 2022)) was utilized to measure environmental conservation value in South Korea. The ECVAM evaluates the legislative and ecological grade from 1 to 5, by synthesizing various environmental aspects of the entire national territory. The legislative ECVAM consists of 62 conservative areas including green belt, while the ecological ECVAM evaluates the potential values for preservation [32]. When the ECVAM grade of a certain raster cell is close to 1, it indicates that an area has higher environmental preservation value and thus lower development possibility [33].

### 2.3. Methods

2.3.1. Research Procedure

Figure 3 illustrates the overall research procedure of the study. It can be largely divided into two parts: (1) development of urban expansion model, and (2) prediction of future urban expansion in South Korea.

First, this study adopted the XAI approach by integrating the XGBoost–SHAP model to predict the urban expansion in Korea. As described in Table 1, the dependent variable was a dummy variable that includes urbanized and non-urbanized area from 2007 to 2019, and the independent variables include socioeconomic, topographic, land-cover, and environmental features. In order to evenly extract samples of urbanized and non-urbanized area across the study area, we rescaled the 10 m × 10 m raster to 50 m and 500 m, respectively, and choose centroid of each raster as study samples. Then, study samples were divided into training and validation dataset for XGBoost–SHAP modeling, which account for 80% and 20% of total samples.

Second, we predicted the future urban expansion of South Korea in 2031 using the constructed XGBoost model. As predictors, a land-cover map in 2019 and corresponding socioeconomic, topographic, land-cover, and environmental features were utilized. Output raster includes the probability of urbanization, from 0 to 1, for all 10 m × 10 m raster cells in the study area.

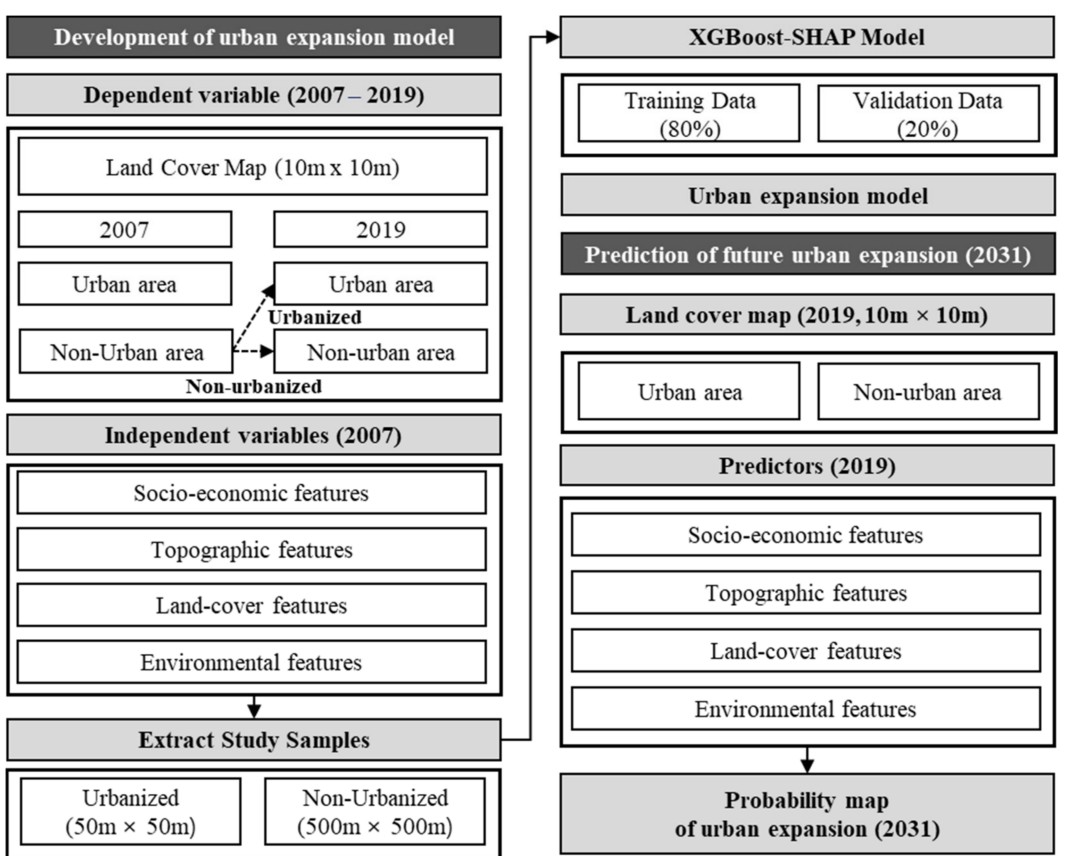

**Figure 3.** Research procedure.

2.3.2. XGBoost–SHAP Model

To develop an urban expansion model, this study combines the eXtreme Gradient Boosting (XGBoost) algorithm and Shapley Additive ExPlanations (SHAP).

The eXtreme Gradient Boosting (XGBoost) algorithm is an open-source library that supports an efficient implementation of gradient-boosted decision trees [34]. Since the package includes efficient linear model solver and tree learning algorithms, users can compute much faster than other existing gradient boosting tools. For the analysis, we used the version 1.5.2.1 of "xgboost" package in R, which was released in February 2022.

The gradient boosting decision tree (GBDT) is an ensemble learning technique that combines a series of weak decision trees to build a strong learner [35]. In this algorithm, each decision tree is trained from the residuals of the previous one, and iteratively constructs a more accurate model until the loss function is minimized. Due to its high predictive precision and ability to deal with both categorical and continuous variables, GBDT has been widely used in various fields of research [36].

For a given training dataset $\{(x_i, y_i)\}_{i=1}^{N}$, the initialized model with constant value is defined as below:

$$f_0(x) = argmin_\gamma \sum_{i=1}^{N} L(y_i, \gamma) \tag{1}$$

where $\gamma$ denotes the constant, $L(y, F(x))$ denotes a differentiable loss function of $\gamma$, and $argmin_\gamma$ indicates the value $\gamma$ that minimizes the function.

For $m$ number of iterations, the negative gradient of the loss function is calculated as

$$g_m(x_i) = -\left[ \frac{\partial L(y_i, f(x_i))}{\partial f(x_i)} \right]_{f=f_{m-1}} \tag{2}$$

Here, $g_m(x_i)$ is calculated by taking a derivative of previous loss function $f_{m-1}(x)$.

Then, a base learner (or weak learner) solves the optimization problem, as follows:

$$\theta_m = argmin_\theta \sum_{i=1}^{N} L(y_i, F_{m-1}(x_i) + \theta t(x; \mu_m)) \tag{3}$$

where $L(y_i, F_{m-1}(x_i) + \theta t(x; \mu_m))$ indicates the loss function on each node $i$.

Lastly, the model is updated as

$$f_m(x) = f_{m-1}(x) + \theta_m t(x; \mu_m) \tag{4}$$

Here, $t(x; \mu_m)$ denotes the selected node and $\theta_m$ denotes the learning rate.

In the XGBoost model, there are several parameters that require to be designated to maximize performance of the model, while preventing overfitting problems [37]. More specifically, the model needs to select the suitable number of iterations, maximum depth, the fraction of observations, and learning rate. In addition, the parameters including 'colsample_bytree', 'alpha', and 'lambda' determine the weights and fitness of the model.

The Shapley Additive exPlanations (SHAP) was first proposed by Lundberg and Lee [38], and has been used to evaluate the relative importance of features in machine learning models. In SHAP, the importance of each independent variable on the model outcome is calculated based on its marginal contribution [39]. For an XGBoost model of group N with n features, the SHAP value $\phi_i$ assigned to each feature $i$ is represented as

$$\phi_i = \sum_{S \in N} \frac{|S|!(n - |S| - 1)!}{n!} [v(S \cup \{i\}) - v(S)] \tag{5}$$

where $S$ represents feature subsets derived from $n$, and $v$ represents the input features within the set $S$.

## 3. Results

### 3.1. Comparison between Urbanized and Non-Urbanized Area

Table 3 summarizes the descriptive statistics of urbanized and non-urbanized area from 2007 to 2019 in South Korea. The number of samples for urbanized and non-urbanized area used in the study were 190,977 and 353,788, respectively. There were significant differences between urbanized and non-urbanized area in terms of socioeconomic, topographic, land-cover, and environmental features.

Regarding the socioeconomic features, the average population density and GRDP per capita were relatively high in urbanized area compared to non-urbanized area. It is not surprising that urban areas tend to expand from densely populated and economically developed metropolitan areas [40–42]. In addition, it implies that urban sprawl is one of the most prevalent types of urban expansion in South Korea.

For topographic features, on the other hand, non-urbanized area showed higher elevation and slope than urbanized area. This is line with the previous studies' findings that the high altitude and slope of a certain land are two of the main influencing factors that hinder development into urban areas [5,43].

In terms of land-cover features, the nearest distance to the majority of land-cover types were shorter in urbanized area, except for forest area. Since residential, commercial, industrial, and transportation areas are largely classified as built-up areas, the probability of new development of a specific area increases as it approaches those land-cover types [44,45].

With regard to environmental features, urbanized area tended to be evaluated as having higher ecological and legislative ECVAM grade, compared to non-urbanized area. Since ECVAM grade indicates the level of conservation of a certain land, a certain raster cell that has higher ECVAM level would be more actively developed [46].

**Table 3.** Comparison between urbanized and non-urbanized area.

|  |  | Total | Urbanized | Non-Urbanized |
|---|---|---|---|---|
| Number of samples | | 544,765 | 190,977 | 353,788 |
| **Socioeconomic features** | Population density (person/km$^2$) | 505.99 | 1176.58 | 144.00 |
| | GRDP per capita (KRW 1,000,000/person) | 23.28 | 27.30 | 21.11 |
| **Topographic features** | Elevation (m) | 200.51 | 82.55 | 264.19 |
| | Slope (°C) | 13.79 | 5.90 | 18.05 |
| **Land-cover features (distance to nearest:)** | Residential area (m) | 528.44 | 269.65 | 668.14 |
| | Commercial area (m) | 1881.78 | 1268.82 | 2212.67 |
| | Industrial area (m) | 2335.20 | 1263.24 | 2913.85 |
| | Transportation area (m) | 687.68 | 340.89 | 874.87 |
| | Agricultural area (m) | 229.90 | 123.97 | 287.09 |
| | Forest area (m) | 118.06 | 197.20 | 75.34 |
| | Grassland area (m) | 74.14 | 51.07 | 86.59 |
| | Wetland area (m) | 79.61 | 48.85 | 96.21 |
| | Bare land area (m) | 166.56 | 130.73 | 185.91 |
| | Water area (m) | 807.71 | 602.59 | 918.43 |
| **Environmental features** | Ecological ECVAM grade | 2.81 | 3.84 | 2.25 |
| | Legislative ECVAM grade | 2.13 | 2.46 | 1.96 |

*3.2. Model Results*

3.2.1. XGBoost Model Results

The optimal XGBoost hyperparameter values chosen for the study are summarized in Table 4. The number of iterations was 100, and the maximum number of tree splits was 6. Of the total, 80% of the samples were used for training, with learning rate of 0.3. To prevent overfitting of the model, 'colsample_bytree', 'alpha', and 'lambda' value were tuned as 1, 0, and 1, respectively, based on the cross-validation.

**Table 4.** Hyperparameter tuning of XGBoost model.

| Parameters | Values |
|---|---|
| Number of iterations | 100 |
| Max depth | 6 |
| Subsample ratio | 0.8 |
| Learning rate | 0.3 |
| Colsample_bytree | 1 |
| Alpha | 0 |
| Lambda | 1 |

Overall accuracy of the XGBoost model developed in the study was 82.54%, where 89,930 raster cells among 108,953 cells were correctly modeled (Table 5). In detail, the ratio that predicted urbanized cell as urbanized, and non-urbanized cell as non-urbanized were 76.99% and 85.53%, respectively. The model better predicts the non-urbanized area compared to the urbanized area.

**Table 5.** Accuracy of XGBoost model.

| | | Predicted | | Total |
|---|---|---|---|---|
| | | Non-Urbanized | Urbanized | |
| Actual | Non-urbanized | 60,524 | 10,241 | 70,765 |
| | Urbanized | 8786 | 29,402 | 38,188 |
| Total | | 69,310 | 39,643 | 108,953 |
| Accuracy (%) | | 85.53% | 76.99% | 82.54% |

### 3.2.2. Factor Importance

Figure 4 illustrates the estimated SHAP values of the study's XGBoost model. It indicates the importance and direction of independent variables in determining if a certain area is likely to be urbanized or not.

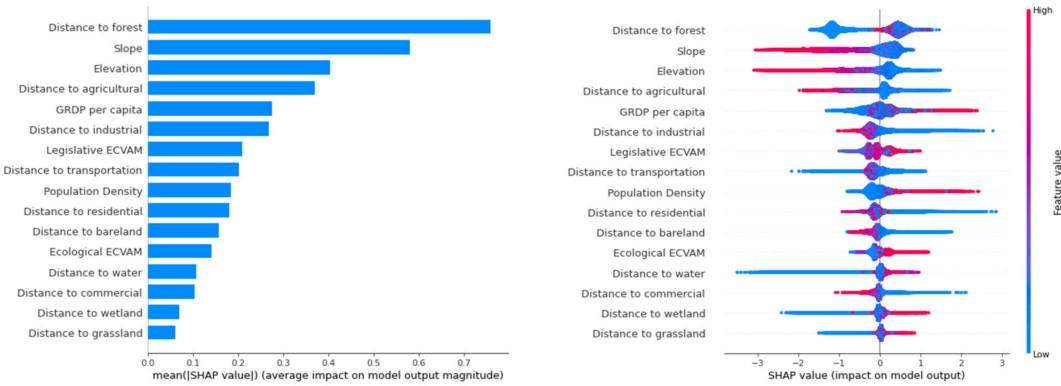

**Figure 4.** Factor importance (SHAP value).

First of all, the distance to nearest forest area was found to be the most influencing factor. The XGBoost model of this study showed that the probability of urbanization increases as the distance to nearest forest area increases. Second, the topographic features, including slope and elevation, also showed relatively high significance to urban expansion. Both factors were negatively associated with the urbanization of a certain area. Those findings suggest that the physical availability of development is the most influencing factor on urban expansion [5,43].

Socioeconomic features were also found to be relatively significant predictors of urban expansion. The GRDP per capita and population density of a certain region tend to accelerate the development of the nearest urban area. It is notable that the economic vitality is a more important factor of urbanization, compared to the population number.

On the other hand, the distance from existing urbanized areas did not significantly affect the level of urban expansion, particularly for residential and commercial areas. Instead, the distance from the nearest industrial and transportation areas was found to be negatively associated with the possibility of urbanization.

In terms of environmental features, the legislative ECVAM grade was a more significant predictor of urban expansion than the ecological ECVAM grade. This outcome makes sense in that the legislative ECVAM grade is based on several legal restrictions of the land development, while the ecological ECVAM grade only recommends consideration for development without coercion [47,48]. As both legislative and ecological ECVAM grades increased, the possibility of urban expansion also increased.

### 3.3. Urban Expansion Prediction

Figure 5 shows the probability map of urban expansion in 2031, based on the land-cover maps of 2019 and XGBoost model developed in the study. The output raster cells

were calculated as 10 m × 10 m units. As a result, the predicted areas of urban expansion at 80%, 90%, and 95% level of probability were 2245.36 km$^2$, 501.74 km$^2$, and 131.31 km$^2$, respectively, which account for 0.57%, 0.12%, and 0.03% of the whole country.

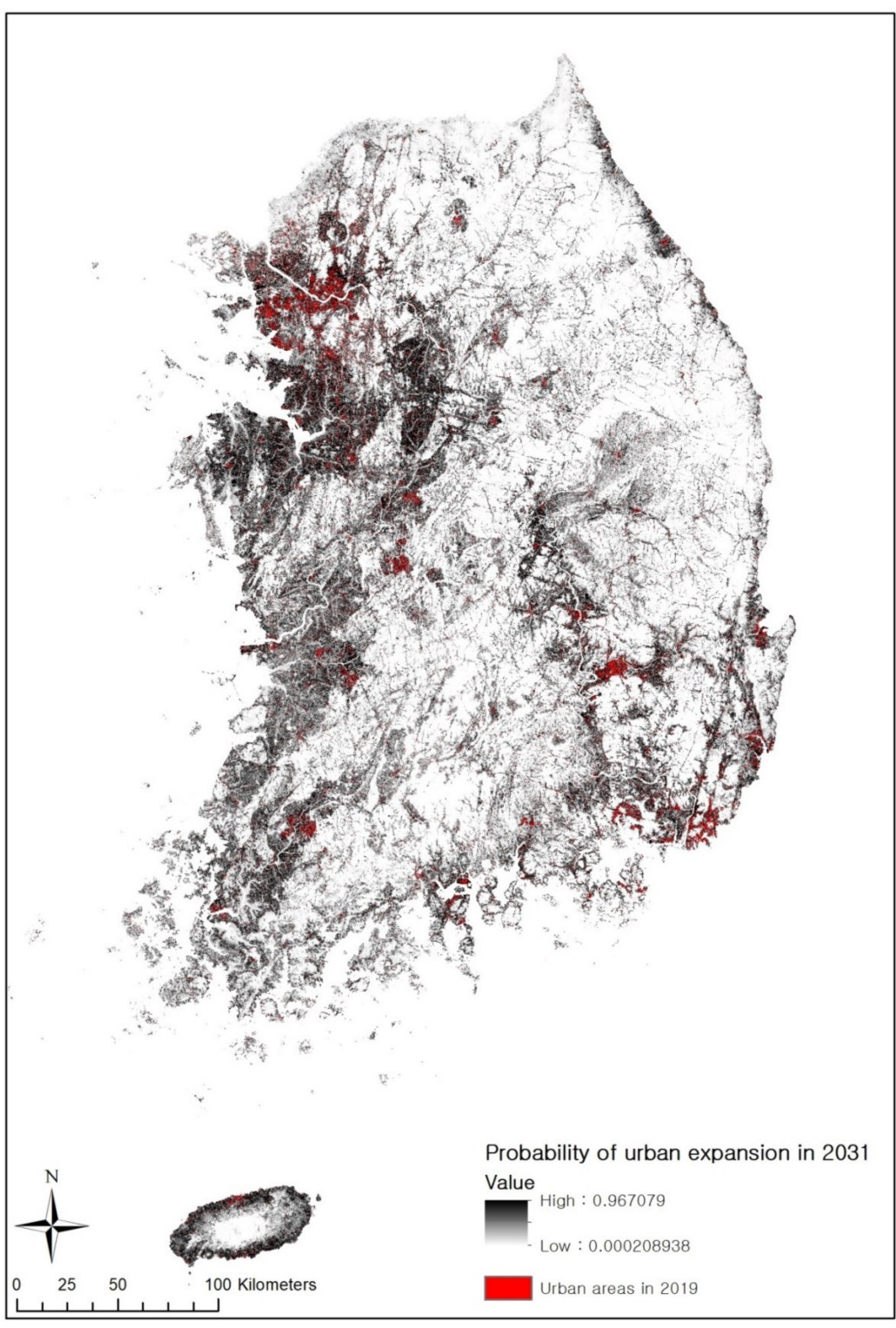

**Figure 5.** Probability map of urban expansion in South Korea (2031).

One of the most noticeable points in this map is that raster grids with a relatively high probability of urban expansion were found to be concentrated in the northwestern part of the country. This seems mainly due to the existence of the Seoul Metropolitan Area (SMA)

within this region, which comprises the city of Seoul, Incheon, and Gyeonggi-do provinces. Those areas account for almost half of the national population and GDP [49,50]. For a similar reason, the raster grids that are adjacent to other metropolitan areas in South Korea, including Busan, Daegu, and Daejeon, showed high probability of being urbanized area.

In addition, the spatial boundaries of expanding urban areas seem to be highly dependent on the geographical and topographical characteristics of the country. The vast majority of raster grids with less than 10% probability of urban expansion are currently covered by the country's mountainous areas with high elevations and slopes. Instead, flat and wide agricultural areas such as paddy fields showed relatively high probability of being urbanized.

## 4. Discussion

In traditional approaches, urbanization was often understood as a linear and physical process, including top-down land-use planning [51]. Recently, however, researchers have examined the nonlinear and socioeconomic properties of urban expansion with advanced modeling algorithms such as machine learning (ML) and artificial intelligence (AI) models [19,52]. While these techniques greatly improved the accuracy of predicting urban expansion, their black-box nature has been pointed out as a major limitation in understanding the relative importance of influencing factors of urbanization [53,54].

From this point of view, our study is novel in the existing literature from several perspectives. First, the study adopted the XAI approach by integrating the XGBoost–SHAP model in predicting the urban expansion in South Korea. It enabled the interpretation of magnitude and direction of influencing factors in predicting the urban expansion, which has not been thoroughly investigated in previous studies using the ML and AI techniques [27,30]. As a result, the vicinity to green area and the existence of harsh topographic environments, such as slope and elevation, were found to be the most influencing factors that prevent the expansion of urbanized areas.

In addition, the study takes a theoretical step forward from previous studies, in that it examines the relative effects of social and economic factors in predicting the urban expansion. More specifically, the study shows that the level of economic development tends to more promote urbanization compared to the density of populations. It complements the existing studies' findings that the urban expansion has been mainly dependent on population growth [55,56]. It is also noteworthy that the legislative ECVAM grade was found to have more significant impact on urban expansion than the ecological grade. This suggests that the environmental regulations on land use can affect the spatial pattern of urban expansion.

Based on the study's findings, we suggest several policy implications for the cities' sustainable development. First, planners and practitioners in the urban planning field need to narrow down the spatial extent of urban expansion based on the geographical and topographical features of the target regions. In other words, scientific judgement on whether a certain area will be urbanized or not should be determined prior to developing strategies for the urban expansion control. Second, the designation of appropriate legal restrictions on development can be effective tools in managing the level of urban expansion. Based on our findings, the authorities can establish site-specific conservative zones to control excessive urbanization. Last, urban planners should be aware that the economic level of a certain city can be an important predictor of urban expansion. In order to prevent the spatial imbalance of urbanization across the country, it is necessary to prepare appropriate measures for economically underdeveloped regions, such as attracting companies or promoting the tourism industry.

## 5. Conclusions

Analyzing and predicting the expansion of urban areas has long been an area of interest in remote sensing and urban study sectors. By adopting XAI modeling, this study developed the urban expansion model and predicted the possibility of urbanization in the

near future. The study's results suggest that the urban expansion tends to be promoted when a certain area is close to economically developed area with gentle topography. In addition, the existence of mountainous area and legislative regulations on land use were found to significantly reduce the possibility of urban expansion. Our findings can contribute to develop cities' effective strategies in managing sustainable urban expansion in the future.

Despite the study's contribution in predicting urban expansion, there is still some room for improvement in future research. First, the XGBoost model has resulted in relatively high accuracy in recent urban studies, particularly for transportation sectors [37,57]; however, the prediction accuracy of urban expansion derived in this study's model was not significantly higher than previous studies that used other methods, including logistic regression and machine learning techniques [7,24]. It seems to reflect the complex nature of urban expansion processes, which is determined by not only socioeconomic and physical environment, but also political and cultural factors [58,59]. To increase the accuracy of predicting urban expansion, researchers are required to consider various aspects of the target areas.

In addition, the study's urban expansion model did not take into account the effects of urban decline, which is currently occurring in many developed countries around the world. In South Korea, for example, the population number has been decreasing in the late 2010s, and small cities are declining in terms of demographic and economic aspects [60]. However, the study predicted that the urban areas tend to continuously expand in 2031, as we developed the urban expansion model using urbanization data from between 2007 and 2019. Future studies need to consider not only factors influencing urban expansion, but also those for urban declines through widening of temporal extent of analysis.

**Author Contributions:** Conceptualization, M.K. and G.K.; methodology, M.K. and G.K.; software, M.K.; validation, M.K.; formal analysis, M.K.; investigation, M.K.; resources, G.K.; data curation, M.K.; writing—original draft preparation, M.K.; writing—review and editing, G.K.; visualization, M.K.; supervision, G.K. project administration, G.K.; funding acquisition, G.K. All authors have read and agreed to the published version of the manuscript.

**Funding:** This work is supported by the National Research Foundation of Korea (NRF) grant funded by the Korea government (MSIT) (NRF-2020R1C1C1013582).

**Institutional Review Board Statement:** Not applicable.

**Informed Consent Statement:** Not applicable.

**Data Availability Statement:** Not applicable.

**Acknowledgments:** This paper is based on the findings of the research project "A Study on Data-based Environment Inequality and Influence Analysis Techniques Using Machine Learning and Spatio-temporal Analysis" (2022-037(R)), which was conducted by the Korea Environment Institute (KEI).

**Conflicts of Interest:** The authors declare no conflict of interest.

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
