# Peer review of "Modeling and Predicting Urban Expansion in South Korea Using Explainable Artificial Intelligence (XAI) Model"

_applsci, doi:10.3390/app12189169_

Round 1

Reviewer 1 Report

The authors have chosen one of the most interesting areas for their paper - that being the use of a remote approach to the development of urban spaces. They mention several methods before presenting their own. The methodology is clear and well written. The diagrams are clear and readable. The weak point is the discussion section. Possibly it would be well if the authors dedicated some space to the German approach to spatial planning i.e. Machine Learning Algorithms for Urban Land Use Planning: A Review, written by Vineet Chaturvedi * and Walter T. de Vries. This will make the discussion more robust, as well as interesting. Also, it would further support the argument expressed by the authors in line 281 "Compared to existing literatures, this study is novel". This might also further enhance the conclusions which are somewhat weak in view of presented analysis,

Author Response

Thank you for your comments. We carefully reviewed your suggested paper and improved discussion part of the study. More specifically, we highlighted the limitations of previous urban expansion modeling techniques such as machine learning (ML) and artificial intelligence (AI) and the strength of XAI approaches in understanding the relative importance of influencing factors of urbanization. (p6, p11-12)

Reviewer 2 Report

The final sentence of the abstract needs to include more specifications e.g. what is the novelty.

For the equations 3, 4 and 5 all the symbols and operators should be explained.

How the it is XAI application, needs more discussion in the main content.

In the conclusion first state your conclusions then draw interpretation towards future work.

Author Response

The final sentence of the abstract needs to include more specifications e.g. what is the novelty.

  • Thank you for your comments. As suggested, we specified more about the novelty of the study in the final sentence of the abstract (p1 line 12-14). Compared to previous studies, this study is novel in that it captures the relative importance of various influencing factors in predicting the urban expansion by integrating the XGBoost model and SHAP values.

For the equations 3, 4 and 5 all the symbols and operators should be explained.

  • We added and double-checked the explanations about all the symbols and operators for all equations in the manuscript (p7).

How the it is XAI application, needs more discussion in the main content.

  • We discussed more about how XAI methods are applied in this study in both theoretical and methodological aspect. More specifically, we highlighted the limitations of previous urban expansion modeling techniques such as machine learning (ML) and artificial intelligence (AI) and the strength of XAI approaches in understanding the relative importance of influencing factors of urbanization. (p11-12)

In the conclusion first state your conclusions then draw interpretation towards future work.

  • Thanks for the comments. We stated a purpose and summary of the study’s key findings in the first part of the conclusion section (p12).

Round 2

Reviewer 1 Report

Accept after minor revisions. Authors however should note that research is done for the further development of reality.

Author Response

Thanks for your comments. We more emphasized about the importance and applicability of our study's findings in the manuscript (p15). Then, we suggested several policy implications in developing strategies for sustainable urban expansion.